# Enhancing Livability in Informal Areas: A Participatory Approach to Improve Urban Microclimate in Outdoor Spaces

Heba Allah Essam E. Khalil [1,*], AbdelKhalek Ibrahim [2], Noheir Elgendy [1] and Nahla Makhlouf [1]

1   Department of Architectural Engineering, Faculty of Engineering, Cairo University, Giza 12613, Egypt;
    noheir@gmail.com (N.E.); nahla.nabil@yahoo.com (N.M.)
2   Faculty of Urban and Regional Planning, Cairo University, Giza 12613, Egypt; abokhalek@yahoo.com
*   Correspondence: hebakhalil@eng.cu.edu.eg

**Abstract:** Urban informalities have shown global unprecedented growth rates in the past 50 years, currently housing around two thirds of Cairenes. As outdoor activities are fundamental to urban life, improving environmental performance of this urban product is essential. This paper investigates microclimate in Cairo's informal areas and how to improve it through low-tech interventions. It aims to identify relevant heat stress counterstrategies, and how they can be promoted among and accepted by residents. Building on previous work regarding an Outdoor Spaces Environmental Performance Assessment OSEPA tool, the researchers identified needed intervention areas within Imbaba informal district. Through an extensive participatory process, the team developed tailored solutions that help improve the urban microclimate using low tech and low-cost interventions. The field work identified vital prerequisites and revealed the necessity of engaging various stakeholders to ensure ownership and applicability. The analysis concludes with an Outdoor Spaces Environmental Performance Improvement OSEPI model for developing solutions to improve urban microclimate within spaces of informal areas and a toolbox for relevant interventions. This serves both as an input in informal areas upgrading projects and designing low to middle-income neighbourhoods. Thus, ensuring environmental justice and counteracting current practices that predominantly ignore environmental issues especially increased heat stress.

**Keywords:** environmental performance; heat stress; informal areas; OSEPI model; outdoor spaces; urban microclimate

## 1. Introduction

The UN Department of Economic and Social Affairs in 2018 reported that 55% the world population lives in urban areas and expected to reach 68% by 2050 [1]. While Mohajerani et al. expects it to reach 61% by 2030 [2]. Urban population rise is associated with an increase in number and size of megacities, those with more than ten million inhabitants, since the second half of the 20th century. There are 23 megacities worldwide, which account for 10% of the world population and cover only 0.2% of Earth's surface [3] and the UN DESA predicts the world to have 43 megacities by 2030 [1]. Although megacities are considered urban centres for economic and industrial activities [3], according to a study conducted by Ningrum, W. [4] in Bangkok, urbanization is claimed to be the main cause for air temperature rise. Most of these megacities are located in the Global South, where plans for air quality and environmental management are lacking. According to Baklanov et al. [3], Cairo was the second largest megacity in Africa and the first in the Middle East and North Africa (MENA) region, in 2011, yet it is currently the largest megacity in Africa with population surpassing 20 million [1]. In cities within such a context, mitigation and adaptation strategies will require working on several levels, one of them is informal settlements [5]. To apply these strategies, different stakeholders such as scientists, communities, urban planners, and decision makers must come together.

Anthropogenic activities, industrial activities, differences in heat capacity of building materials [3] and increased surface area from added building and manmade materials [6] make megacities prone to increased levels of air pollution, higher temperature, poor air quality levels and affect air circulation and microclimate [7]. The rise of air temperature of urban areas more than the surrounding rural regions by several degrees, up to 15 °C is a phenomenon called Urban Heat Island (UHI) [2,3,6,8]. UHI is a result of manmade materials having lower albedo, especially materials used for external building facades and roads [9], than other surrounding natural surfaces that hold water, e.g., vegetation. Subsequently, these materials are more able to absorb surrounding solar radiation causing the increase in air temperature [4]. Furthermore, UHI is linked also to the use of materials capable of storing shortwave radiations during daytime and then releasing them as longwave radiation in night-time [9]. Moreover, Ningrum, W. [4] proved using simulation software that distance between building plays more key role than building type in affecting urban microclimate. This is in accordance with Rad et al. [7], who claim that the ratio between building heights and street width has a positive correlation affecting the increase of UHI as increased building heights could prevent air movement and slowdown the nocturnal release of heat stored during daytime. UHI further affects public health and wellbeing, causing heat related human health problems, and human discomfort [3,9], which could result in mortality and morbidity [10]. Furthermore, Palme et al. [11] demonstrated that UHI could contribute to 15% to 200% increase in building energy demand using building performance simulation. Thus, measures to mitigate UHI are not only important to improve outdoor thermal comfort on the urban scale but further it affects building energy consumption. Megacities in warm climates are more affected by such consequences [10]. Therefore, having an action framework with practical steps for implementation of mitigation as well as adaptation strategies is important.

Urban life is predominantly defined by the quality of outdoor spaces urban dwellers enjoy. In the many indices to measure Quality of life and Livability, quality of public spaces has been an implicit indicator [12–14]. For these indices, these spaces support community life and provide recreational opportunities. Other indices have identified quality of open spaces as a fundamental indicator of quality of life as the City Prosperity Index [15]. In the Sustainable Development Goals, SDG 11 "Making cities inclusive, safe, resilient and sustainable" gives special focus to open spaces within the concept of inclusiveness [16]. Additionally, the New Urban Agenda emphasizes the importance of addressing quality of open spaces within efforts for improving livability especially to the urban poor [17]. This issue becomes more complex within open spaces of informal areas where spaces are scarce and with increased vulnerability to climate change and increased heat stress. Accordingly, this paper investigates possible responsive solutions to improve the resilience of outdoor spaces to increased heat stress. It builds on extensive fieldwork and participatory engagement from multiple stakeholders, in "Imbaba" one of the densest informal areas in Cairo, to propose appropriate interventions that would improve the environmental performance of outdoor spaces in such contexts. It proposes a model for identifying interventions that are low tech and low cost responsive to the needs and characteristics of informal/unplanned areas and could be extended to spaces of other low-middle income areas.

## 2. Microclimate in Informal Areas

### 2.1. Increased Heat Stress in Informal Areas

The vulnerability of developing countries to heat stress and microclimate implications is far more severe than that of developed world as population growth is expected to be almost doubled in the former, between 2005 and 2030 [10]. Leal Filho et al. [10] concluded through an analysis of how cities in the southern and northern hemispheres are vulnerable to UHI, that strategies need to be more context-specific for each city in order to make urban areas more resilient to UHI. Although both sprawled and compact cities exhibit UHI [2], disperse urban form has less thermal stress over city centres than compact urban form [18]. As informal areas are compact urban forms which do not comply with planning and

development regulations [18], they have a positive association with UHI, increasing Land Surface Temperature (LST) and thus are vulnerable to heat stress [18]. It is vital to note that three billion people are expected to live in slum areas by 2050 [5]. Within informal area, the importance of outdoor space is unsurmountable given the extremely small residential units and over crowdedness, outdoor space becomes an extension to the normal indoor activities. Additionally, inhabitants of informal areas very often enjoy a strong sense of community and social life [19]. Interestingly, people in these areas are the exact manifestation of the right to the city as outlined by Lefebvre [20]. In fact, their practices far exceed just the right to participate in decision making regarding what happens in their spaces, as they are the ones who produced them in the first place through informal development. Within completely developed areas, they retain their right to appropriate these spaces according to their needs, which change even with the same day. In such contexts, one can clearly observe a full demonstration of what Purcell [21] argued as what the right to the city is about 'right to live in, play in, work in, represent, characterize, and occupy urban space...'. Space users exhibit what Harvey [22] emphasizes regarding the public space becoming a venue for diverse political collectiveness as people participate in public life. However, with such intensity of engagement and existence, spaces become contested among different vested interests that necessitate a careful approach from designers and planners seeking to intervene in such contexts. It is crucial to understand and hence respect the existing social structures that govern the space. Although, many interventions could be implemented from a conceptual point of view, the local socio-economic network shape intervention possibilities.

A study conducted, by Scott et al. [6], in Nairobi at three different informal settlements showed that temperature measured throughout day and night are higher in those areas more than city centre and other surrounding formal/non-slum areas. Another study by Mehrotra et al. [18] analyzed the relationship between slum housing and Surface Urban Heat Island (SUHI) supported by spatial-statistical analysis in the city of Mumbai. Although slum pockets within Mumbai provide an affordable housing option for migrants, they pose an overload on not only infrastructure and wellbeing but also on the urban environment contributing to increase UHI [18]. The study confirmed a strong link between Slum Urban Form (SUF) and LST, and that the contribution of SUF to increase SUHI is more than formal housing areas. This could be reasoned by the large roof surface area of highly dense low-rise housing which contributes to increased heat input to the surrounding.

An approach to improve livability in informal areas could include the improvement of urban microclimate within their outdoor spaces. With the current, and further expected, increase in heat stress induced by climate change and urban activities, it becomes crucial to address the issue especially in marginalized areas where vulnerability is at its maximum. Taking Cairo as an example for megacities with continuously growing informal areas, the data shows that increased heat is expected even under the best climate scenarios [23,24]. In August 2015, Cairo witnessed a sever heat wave reaching an abnormal record 49 °C temperature for 10 days [23,25]. This resulted in several deaths and people being hospitalized due to heat stress [25]. In that sense, it becomes imperative to study the environmental performance of outdoor spaces within informal areas of cities that are vulnerable to heat stress and develop appropriate solutions.

The study of Cairo's urban climate has revealed the extreme high accumulation of heat in most of the city [26]. According to the same study, all informal areas within Cairo are subject to accumulation of heat within their urban structures. This is due to the extreme compactness of buildings, lack of vegetation and extensive use of low albedo materials including asphalt and red bricks (without any plastering or painting). Within such structure, it is crucial to develop solutions that respond to the local context and capacities.

### 2.2. Potential Heat Interventions

Many researchers have identified multiple possible interventions that could be utilized to improve the microclimate [2,8,9,27–30]. Main interventions include façade shading,

shading of the spaces, using materials with high albedo as white roofs and light colours painting and minimizing the use of asphalt [2]. In addition, the extensive use of trees, green walls and green roofs can significantly improve the situation [8,9]. Aflaki et al. [9] demonstrated that urban vegetation can significantly reduce air temperature and mean radiant temperature by 4 °C and 4.5 °C, respectively. Finally, the presence of water bodies features an improved thermal comfort in outdoor spaces. Although an important intervention is to promote ventilation [31], however it is more suitable for new areas than redesigning existing spaces that are highly dense. Conditions of each city are not alike, and solutions cannot be relocated or duplicated [5,18], therefore, heat mitigation measures must be tailored for context-specific conditions [5,10]. The acceptability and appropriateness of these tools to local knowledge and affordability remain undefined. Hence, this paper aims to fill this gap through proposing a pilot project in two open spaces in Imbaba, an informal area within Cairo.

According to the literature previously discussed, this research identified four domains of concern when designing locally responsive interventions to improve environmental performance of outdoor spaces as illustrated in Figure 1.

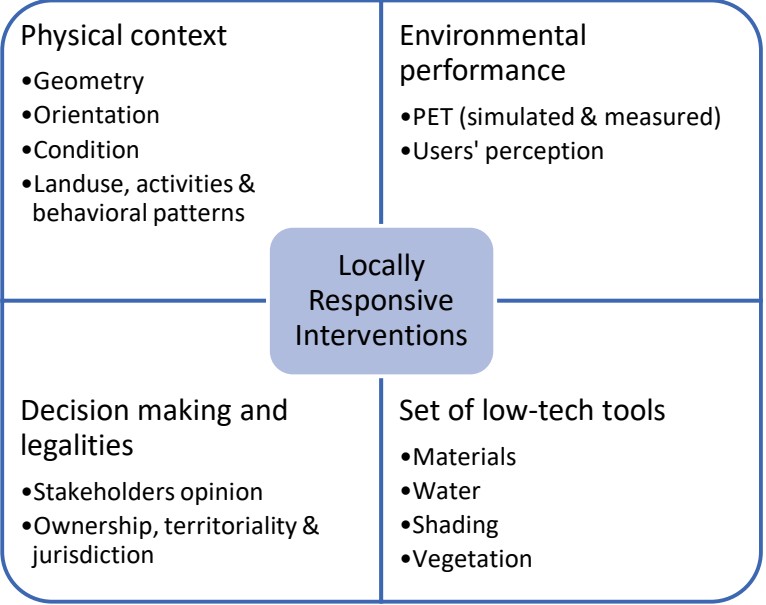

**Figure 1.** The four domains of locally responsive interventions.

These four domains consist of four main constituents; first is the physical context specifically regarding urban geometry, street orientation, land use, building conditions, prevailing activities, and behavioural patterns. The second constituent is the environmental performance, and it has twofold; calculating the Physiological Equivalent Temperature (PET) (using simulation tools as Envi-Met) and investigating the users' perception regarding environmental comfort. The third is concerned with decision-making and legalities, focusing on investigating various stakeholders' opinions. Furthermore, it includes identifying ownerships and territorialities of various users to ensure stakeholders' participation, suitability, and sustainability of interventions. Additionally, it is also vital to identify the legalities of dealing with public space and public ownership and approaches to acquiring governmental approvals. The fourth and the last constituent is the set of low-tech tools. These tools are concerned with four categories. (a) Dealing with materials that are higher in albedo as light colours painting and minimizing the use of asphalt. (b) Water is an essential element to integrate into outdoor spaces however, not always appropriate in extreme dense situations. (c) Shading is especially important whether façade shading, shading of the spaces with wooden, or fabric shades and/or trees. (d) Vegetation increases comfort and reduces UHI through the extensive use of trees, green walls, and green roofs.

## 3. Materials and Methods

This paper adopted a deductive approach within the grounded theory framework. It relied on empirical inquiry to generate relative experience building on piloting and field work to extract an action model for improving environmental performance of outdoor spaces within compact neighbourhoods and especially in informal areas of the Global South. Additionally, the work heavily relied on participatory action research approach to shape the research questions, investigations, data gathering and decision making.

To develop a model for open space environmental performance improvement along with relevant interventions, this study first built on previous investigation in assessing the environmental performance of outdoor spaces. The open space environmental performance assessment OSEPA tool previously developed by the authors combined an urban survey of the space (use, dimensions, scale and materials); socio-economic survey with 600 respondents: 300 residents, 150 shop owners and street vendors, and 150 passers-by (regarding users' behaviour, perception and acceptability/applicability of potential interventions) and activity mapping (during different times of the day and different days of the week); actual environmental measurements (surface temp, air temp, relative humidity (RH) and wind speed); and microclimate simulation using ENVI-met (relying mainly on the PET) to identify hotspots [26]. Khalil et.al. [26] displays more details of the methods, results, and analysis for the identification of different hotspots within the study area.

Second, to develop a responsive and comprehensive intervention plan the team adopted a participatory approach throughout the process to ensure the applicability of the proposals as shown in Figure 2 and detailed below in ten steps.

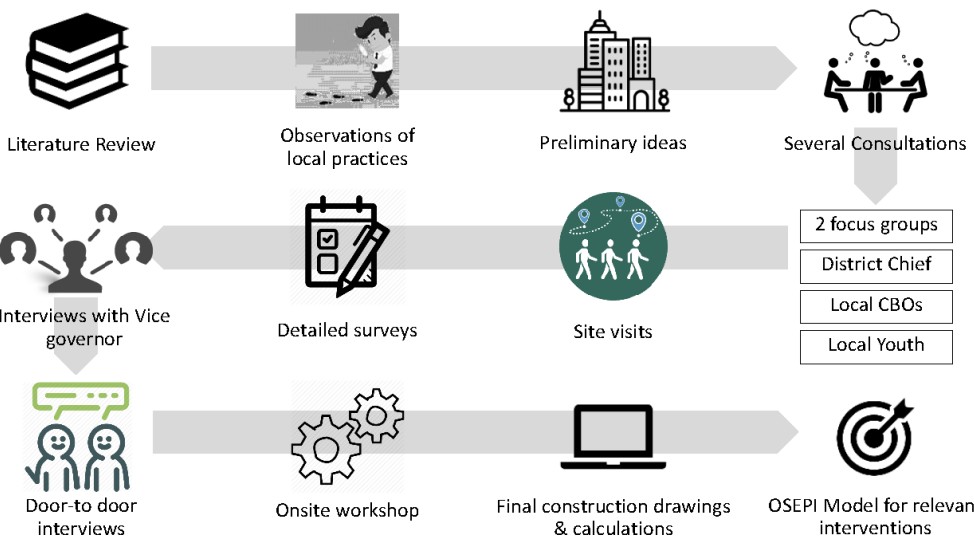

**Figure 2.** Methodology of the adopted participatory approach.

First, relevant intervention domains were identified and extracted from the literature. Second, observation of local practices was conducted through several site visits and transect walks. Third, accordingly a set of intervention ideas were developed for the previously identified hotspots taking into consideration the results of the socio-economic survey conducted previously regarding acceptability/applicability of potential interventions. Fourth, several consultations were conducted with various related stakeholders. These consultations included: (a), two focus groups discussions with some residents. One was conducted with around twenty men in a local café overlooking hotspot 3 (see Figure 3). The group was mostly residents in addition to a few street vendors. The second was conducted at the premise of a deep-rooted Community Based Organization (CBO) located nearby with a group of 15 females living in the area. (b), the district chief was consulted regarding the identified hotspots, proposals, and relevant logistics. (c), local CBOs were contacted regarding the project, hence identifying a CBO coalition with ten local CBOs working together in

Imbaba district. (d), two meetings were held with CBOs coalition in addition to a group of local youth residents within the premise of local community hall located within the district. Fifth, the team conducted several weekly site visits to identify initial views regarding the proposed interventions. Sixth, based on the conceptual ideas, the research team with the help of a group of students conducted a detailed survey of the intended building facades and spaces in the identified hotspots. This included documenting the exact dimensions and materials of different facades and producing relevant drawings. Moreover, the exact dimensions and condition of the two spaces (market space and birds market space) were documented. This detailed survey extended to include sewage systems both within the identified buildings and open spaces. Seventh, the team conducted further interviews with related municipalities. The team reached out to the Governor and Vice Governor to acquire political support and ensure the legality of interventions. Eighth, to ensure an inclusive process, the team conducted door-to-door interviews with seventy-five residents of the two blocks in hotspot 2 suggested for interventions. Interviews investigated residents' opinion about the proposed intervention, to paint their building facades. These interviews were conducted by a group of architecture students in collaboration with social service students supervised by the local CBO. Additionally, interviews were also conducted with a random sample of street vendors. The team used flyers and visual cards to introduce themselves, their work, and the façade designs. Ninth, a workshop was conducted on site where the team used power point presentations and 3D physical models to explain and discuss both the interventions and process with the residents. This was to overcome any shortage in previous meetings or activities. This workshop had the participation of various residents and space users including women and children. Tenth, the team developed an updated set of construction drawings for the suggested interventions according to the outputs of all the participatory process. Accordingly, the authors developed a model to improving environmental performance of outdoor spaces (OSEPI model) in informal areas and a toolbox of relevant possible interventions. Baklanov et al. [3] stressed the need for multi-scale integrated models to be able to deal with challenges related to megacities. Thus, it can benefit decision makers, urban planners, architects, and local community to inspire them with solutions to existing local microclimate challenges.

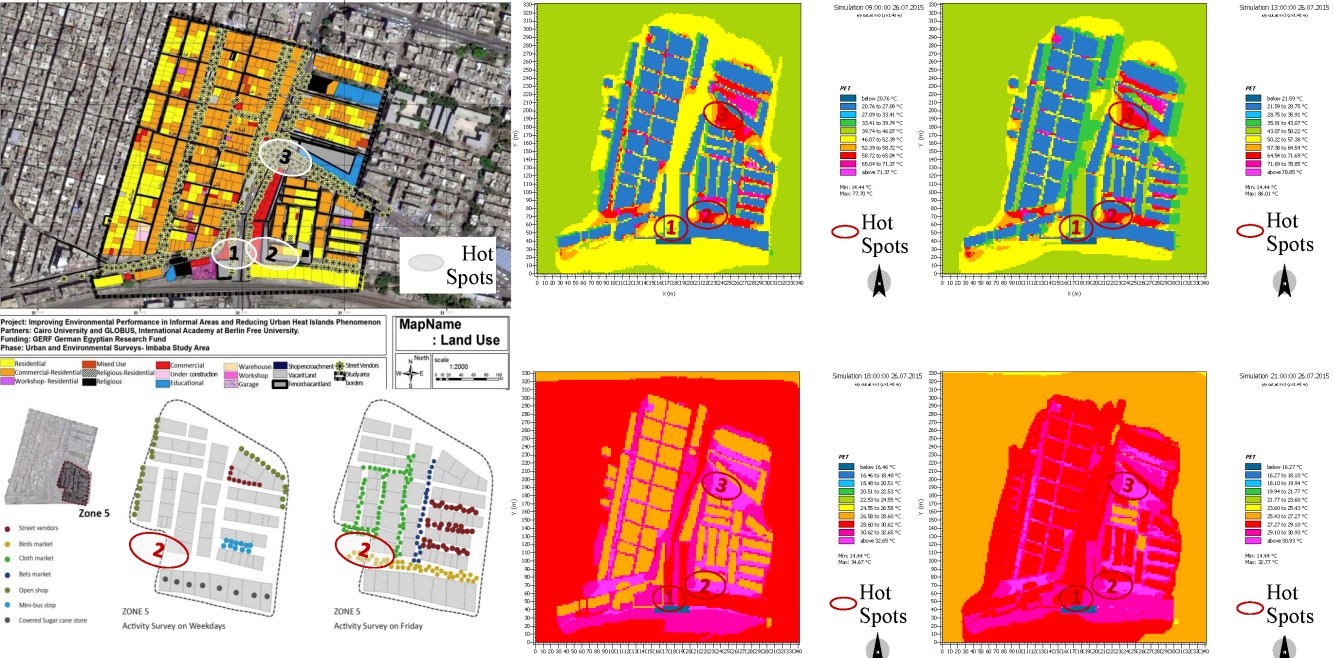

**Figure 3.** The three hotspots as revealed by the Envi-met PET modelling throughout the day (**middle and right**) and compared to the actual use (**upper left**) and activity mapping throughout the week (**lower left**), [26].

## 4. Case Study: Open Spaces in Imbaba, Cairo

Few studies have investigated Imbaba as one of Cairo's densest informal areas [32–37]. Imbaba, previously agricultural land, was illegally subdivided and built to provide affordable housing throughout the past 60 years to reach almost one million inhabitants with up to 1900 persons/ha. A diversity of social and economic strata lives in this area, but mostly inhabitants could be considered of low middle income. In addition, buildings are mostly of mixed use (mainly commercial/residential) with average height of 4–5 floors with some buildings reaching 10–12 floors on the main streets. Almost all buildings are built of skeleton structure with concrete and bricks.

The analysis of thermal sensations using questionnaires and observations with parallel measurements can provide crucial hints for urban planners. Moreover, this analysis can aid in the design of open spaces, where the space configuration influences the microclimatic conditions and thermal sensations considerably. This study and analysis not only relate to the thermal physical but the social environment as well. Within the global calls for new urbanity, the need becomes increasingly crucial to revitalize cities and their public spaces. Hence, considering and improving the environmental conditions and thermal comfort, becomes of vital importance to increase the use of outdoor spaces as a vehicle to revitalize cities. For this study, an area with concentration of activities and mixed uses was selected for microclimate analysis. The area is also a vital entrance to Imbaba housing one of its famous markets (Al-Mounira market).

The assessment of the urban microclimate in the study area as explained in Khalil et al. [26] has identified a few hotspots throughout day and night. Specifically, through the questionnaires, interviews, activity mapping, observations, measurements and modeling, three spots were of special interest as shown in Figure 3. These spots vary in use as shown in Figure 4. Spot one is a vehicular entrance (through an underpass) to the area with continuous congested traffic. Spot two has high solar exposure and is used twice a week as a poultry market, whereas spot three is a wide street used as a market especially after 2 p.m. when employees are on their way home from work. This space specially thrives in summertime when the market extends to the entire day to take advantage of available customers and provide related needs.

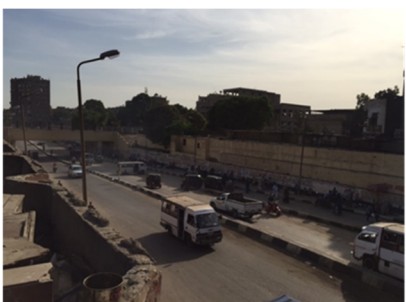
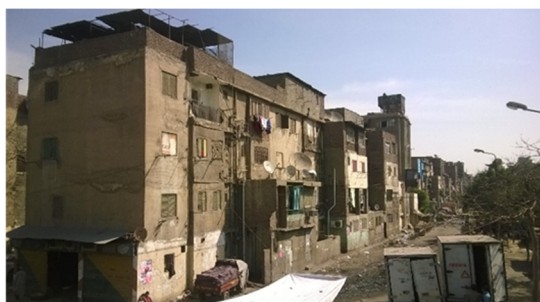
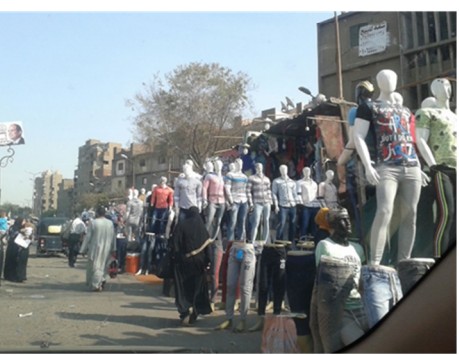
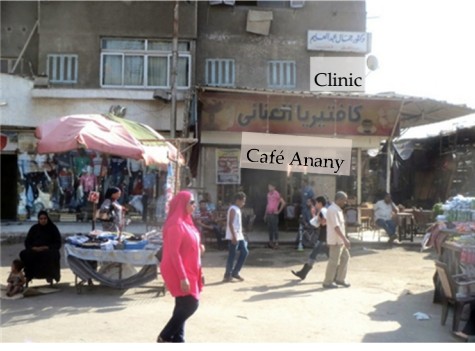

**Figure 4.** Identified hotspots: hotspot 1: the underpass (**top left**), hotspot 2: the birds' market (**top right**), hotspot 3: the market (**bottom**).

## 5. Results

### 5.1. Investigating Current Local Practices

The local context has provided the team with an array of innovative, responsive solutions that the inhabitants have devised to improve their comfort and adapt to increased heat stress. These interventions included space shading using foldable fabric to accommodate certain activities as midday prayers. Within the market space, the use of umbrellas was common along with providing the movable stalls with retractable wooden/or metal shading. Some trees were found in the neighbourhood showing that people are taking diligent care of them even with the complete eradication of previous agricultural land. In addition, the team was able to spot two green walls, which were made of climbing plants, a practice now scarcely found in Cairo. Another widespread practice was installing fenestrations or curtains on windows and balconies, which serve both privacy and thermal comfort purposes as shown in Figure 5. Interestingly, watering the street is a frequent practice, where shop owners or street vendors sprinkle water to reduce dust and cool off the temperature, especially in the afternoon.

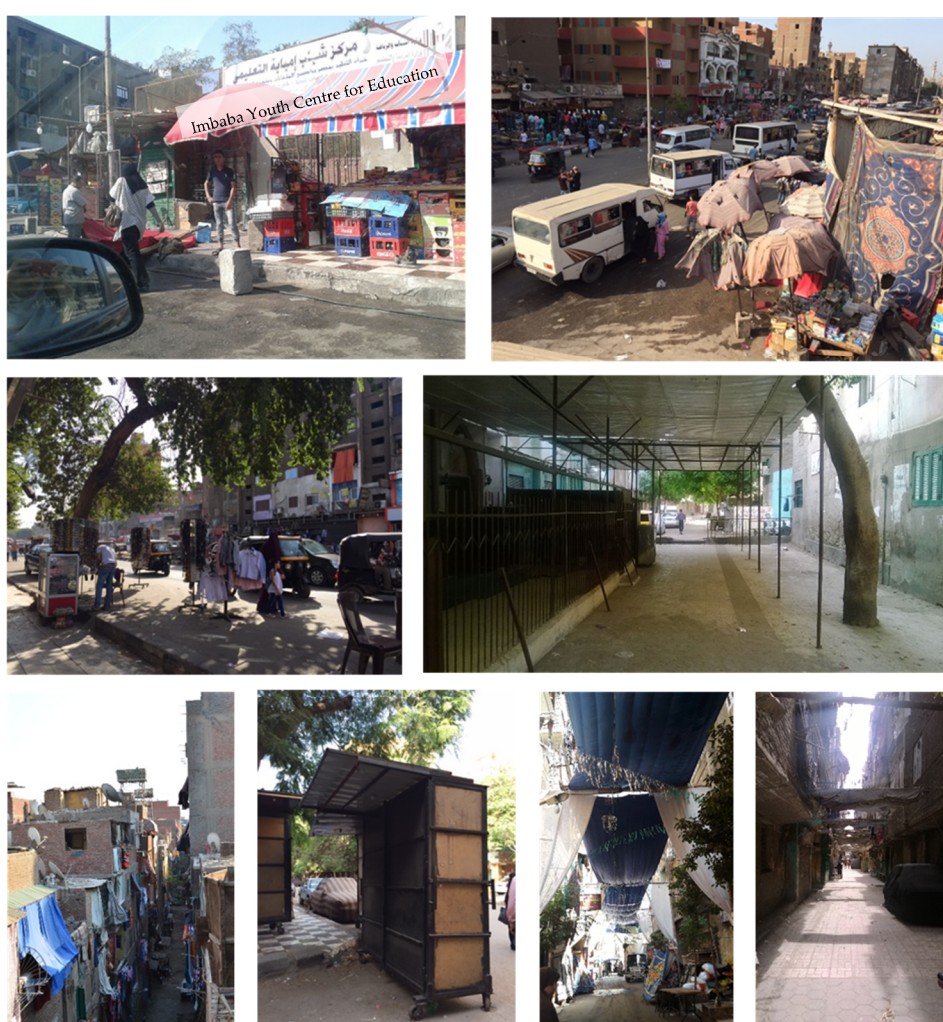

**Figure 5.** Examples of local practices to cope with heat stress and reduce sun exposure.

Concerning the potential interventions related to literature and best practices, a few interventions were identified as not suitable to the context of the study including introducing water features in open spaces and green roofs especially on residential buildings.

### 5.2. The Preliminary Proposals

Based on the results of preliminary focus groups, surveys, questionnaires, and observations of local practices, the set of proposed interventions for the identified hotspots included painting facades with light colours, improving shading and introducing trees and greenery. In addition, investigating the possibility of green roof in the adjacent fish market building was also proposed. The details of the proposed interventions were as follows:

Hotspot 1: The underpass where the proposed intervention is to plant some tress in the middle island and to plant some climbing plants on the rear façade of the fish market building that defines the East boundary of the space.

Hotspot 2: Fish market and Birds market space where the proposed interventions included painting the south façade at the birds' market, introducing green walls, and planting some trees in the market space. In addition, painting the facades overlooking the fish market and installing a green roof on top of the fish market building.

Hotspot 3: Street vendors and market space where the proposed intervention is to plant trees, extend shading from building facades and provide some seats and trash bins to improve the overall quality of the space.

### 5.3. Stakeholders Consultation

Results of the several conducted consultations with various related stakeholders were as follows. The focus groups conducted with groups of men and women revealed the participants refusal for green walls as they encourage the presence of insects and rodents. Moreover, they do not prefer trees too near the facades as they help in dust accumulation inside their apartments.

Consultation with the district chief revealed the impracticality of the interventions in hotspot 1 (the under pass) due to limited available space. Interventions with facades and tree planting were to be supported as long as no trespassing occurs on the public space by buildings extensions. Furthermore, planting fixed trees were encouraged; to prevent them being stolen, even though the initial idea was to plant movable heavy trees so as not to provoke public authorities and public territoriality. Importantly, sheds protruding from buildings are considered (by the municipality) encouraging to encroachments and future building extensions, hence were not acceptable. The district chief also discussed the difficulty of implementing a green roof on top of the fish market building as the building is owned by the district but operated by the fish vendors. Such situation could not facilitate easily roof planting and caring of the crops as well as marketing and managing revenues. The two meetings with the CBOs and local youth as well as the several weekly site visits have further refined the proposals and the development of detailed solutions.

### 5.4. Detailed Solutions

Based on the conceptual ideas and the feedback from the stakeholders first round of consultation, buildings of hotspot 2 and spaces of hotspots 2 and 3 were surveyed to develop detailed solutions based on real conditions. This survey has highlighted a critical issue regarding the facades' deteriorated condition due to leakages from the sewage pipes. Moreover, the deteriorated condition of the sewage manholes in both spaces and the almost continuous contamination with wastewater as shown in Figure 6 made the proposed interventions look similar to a superficial facelift if carried out without properly addressing the sewage system condition.

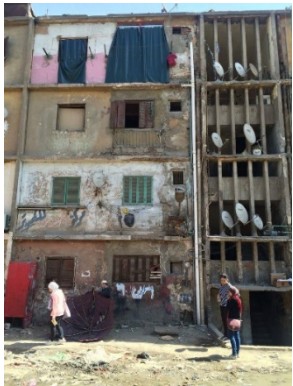
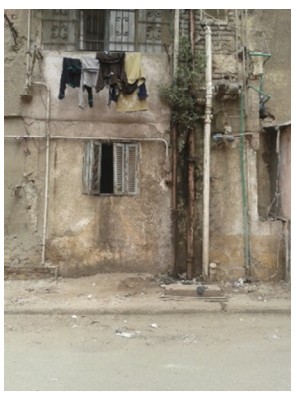
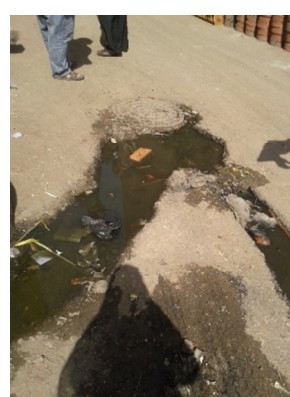

**Figure 6.** Deteriorated sewage pipes and network.

Thus, the team decided to document and assess the current situation of the sewage system on both the space level and the building level as an additional activity to the project. A comprehensive plan was developed accordingly to address the issue and ensure the sustainability of the proposed interventions once implemented. The plan was to replace all deteriorated pipes, clean all manholes, and replace the blocked ones.

The team based on local situation, relevant technologies and worldwide experiences developed alternatives for the refurbishment of the elevations. The alternatives varied in approach and colour scheme; however, the main parameter was to use light colour painting as high albedo instead of low albedo paintings and improve the PET. Figure 7 shows one of the proposals for façade treatment and vegetation in the space of hotspot 2.

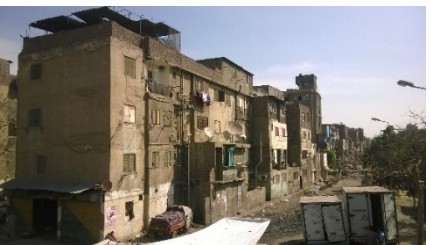
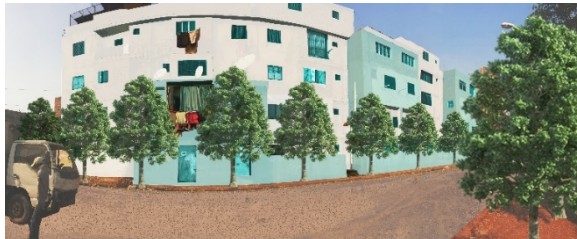

**Figure 7.** The South Facade in hotspot 2 showing the current condition and the proposed intervention.

Moreover, the two spaces for spots two and three were documented and redesigned to incorporate trees and some shading. The main parameter here was to ensure shading all day and ensure that the space remains flexible and maintain its capacity to accommodate for the needs of the shop owners, street vendors, inhabitants, and passers-by. Figure 8 shows the proposed interventions for the birds' market space (hotspot 2) and the market space (hotspot 3).

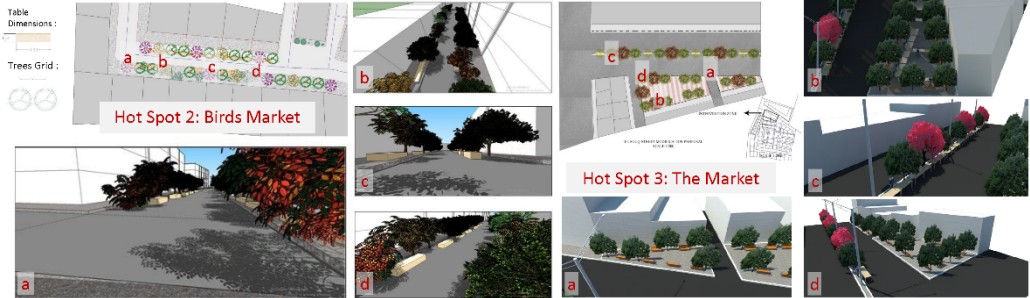

**Figure 8.** Design proposals for spaces of hotspot two (**left**) and hotspot three (**right**) with trees, benches and tile flooring.

*5.5. Stakeholders' Refinements: Interviews with Local Residents, Street Vendors and Municipality*

The door-to-door interviews with the residents and the random interviews with street vendors asked them to choose one of the developed interventions for façade colour and design. A set of visual cards aided the team to help the residents to visualize the proposed interventions. The interviews results showed 65% inclination towards using light green colours in the facades and approving the proposed vegetation and space furniture. Additionally, the consultation with the vice governor has provided political support to the project and interventions and test relevant legalities. The process has proven to be effective to mobilize efforts and resources as the vice governor facilitated the provision of trees and support in improving the sewage system within the streets. It is important to note that through the entire process, the team was accompanied by the leading CBO to ensure partnership and facilitate access to the local community. In addition, the participation of youth from the area, trained and supervised by the CBO, in campaigning to the project have proven highly effective and engaging.

Finally, during the onsite workshop, as the team explained and discussed both the interventions and process to the inhabitants using power point presentation and 3D physical models, participants were highly active and engaged. They decided to form a committee with representatives to facilitate any obstacles during implementation and to collect money for future maintenance and up keeping of foreseen interventions.

The output of this process was developed into execution documents, respective quantities and required budget, and then was commissioned to a local contractor. The estimated cost was 25,000 $ including material and workmanship. The cost of the upgrading of the sewage system was excluded from this budget as the work was supposed to be implemented by the municipality partly with donations from some of the residents. However, the economic reforms that took place in November 2016 in Egypt and the consequent doubling of prices hindered the implementation of the project under the already secured budgets. Hence, the project was not realized.

*5.6. The Proposed Outdoor Spaces Environmental Performance Improvement OSEPI Model*

The pilot study has resulted in developing an action framework to intervene in open spaces for improving environmental performance that addresses the four domains for intervention identified in the literature. The proposed Outdoor Spaces Environmental Performance Improvement OSEPI model for informal areas and similar contexts proposes a set of steps to go through to reach a set of tailored affordable interventions as shown in Figure 9. These steps are as follows:

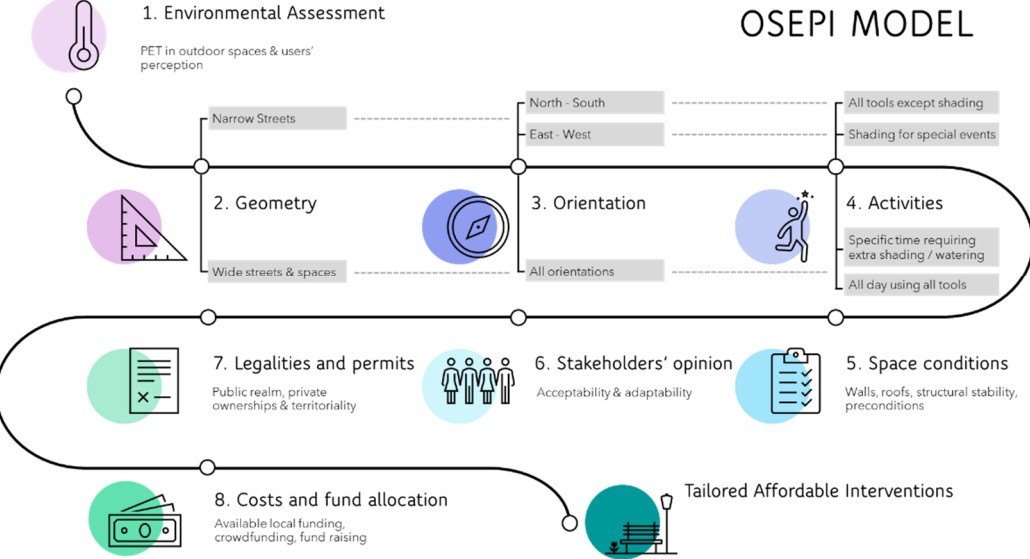

**Figure 9.** Outdoor Spaces Environmental Performance Improvement OSEPI Model in informal areas.

1.  Environmental assessment of the space under investigation. According to the space environmental performance, either the space is considered adequate or further investigation should be conducted in case of increased heat stress.
2.  In case the space suffers from heat stress, the 3D geometry of the space should be studied. In case of wide streets and spaces, shading, vegetation and/or any of the low-tech tools mentioned in Section 5 become essential. The interventions needed in narrow streets are fewer and could be temporal.
3.  Afterwards, orientation should be identified. As specifically, in narrow spaces, shading will only be needed in east –west streets.
4.  Investigating the activities performed in the space (through activity mapping throughout the day and the week) is essential to determine the necessity of interventions and whether they are permanent or seasonal, fixed, or movable.
5.  Hence an investigation and documentation of the space condition is essential with regards to walls, roofs, structural stability, etc. This is carried out to identify any needed preconditions before implementing the designated interventions.
6.  Seeking stakeholders' opinion (through questionnaires, interviews and focus groups) regarding the acceptability of the interventions that respond to the environmental conditions comes next. Their opinion is essential to ensure ownership, appropriateness, and sustainability. Some interventions might need some adaptation to local needs and aspirations.
7.  Ensuring the legalities of the interventions is essential when dealing with public realm and territorialities of various groups and individuals. In addition, acquiring the legal permits from the municipality gives the project stability and legitimacy.
8.  Calculating the required costs comes last and ensuring the availability of funds whether through local inhabitants or interested civil society. Appropriate fund-raising campaigns may be essential at this stage to provide the costs of needed interventions.

## 6. Discussion

Throughout the entire process, the researchers have intensively investigated the issue of environmental performance of outdoor spaces through various methods including theoretical background, similar projects, and fieldwork (urban surveys, socio-environmental surveys, meteorological measurements, and simulation). In addition, a profound participatory process has been devised to reach a locally responsive set of interventions. This process has assisted in formulating a model that can be utilized in similar contexts to improve the urban microclimate using low-tech interventions

The investigation of possible interventions, as suggested in the literature, within the case study has highlighted a few aspects for consideration. First, although water features can improve thermal comfort in outdoor spaces, it would be difficult to introduce this in the area especially within dusty streets and spaces. Moreover, a previous study in the area [32] revealed that the area has a water shortage problem, which would hinder the execution of such intervention along with the recent increase (June 2018) in water prices which would drastically affect affordability. Thus, even local practices of watering the streets have started to become limited. Second, the careful investigation of building roofs revealed the difficulties to implement rooftop planting as most buildings have additions with light materials that take up the roof and cannot accommodate any planting activities on them as shown in Figure 10.

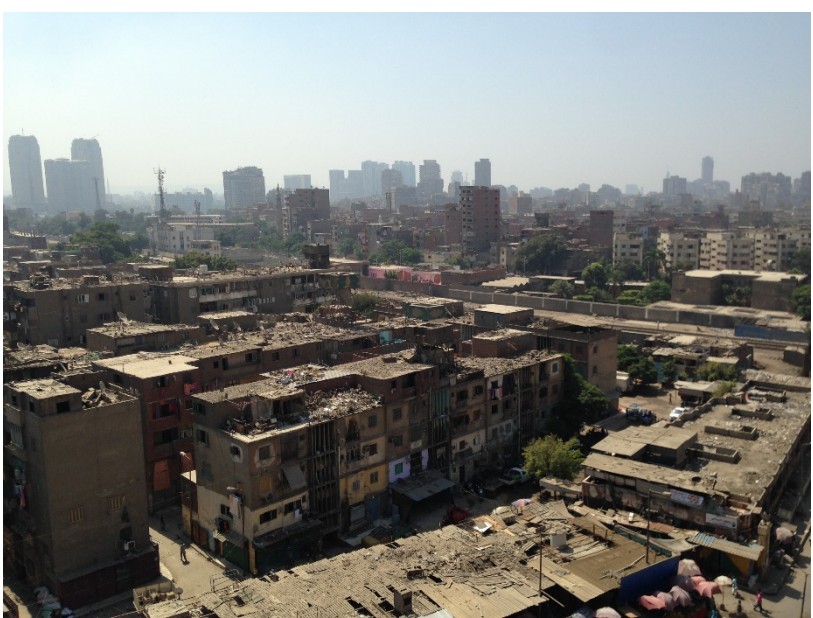

**Figure 10.** Many buildings have additions on the roof built mostly with light weight structures.

Other buildings in surrounding areas might provide better opportunities for such interventions. However, any rooftop planting success relies mainly on developing an appealing business model; meaning that it will produce products that can be marketed nearby. This was apparent in pilot projects implemented by the German Development Cooperation (GIZ) in similar informal districts [38–40]. However, since the available grey water treatment is not suitable for irrigating these produces, thus it will rely on potable water, which may accentuate the water problem in the area. In conclusion, this area of water treatment and reusing needs further investigations and research that goes beyond the scope of this current research. However, there are a couple of promising research efforts undergoing that can provide adequate solutions for this issue [40,41]. Alternatively, roofs can be painted in white/light colour to reduce contribution towards UHI.

The various consultation sessions have revealed different issues that challenge interventions promoted by the literature. The community refusal of climbing trees and green walls reveals the importance of consulting local knowledge that otherwise would go unidentified. Additionally, consultation with the local municipality has revealed challenges of managing public assets when expected revenues would be generated as the case of expected revenues from productive green roofs. Such situations could hinder the application of beneficial interventions where there are no current regulations to govern similar situations. Additionally, although extending temporary vertical covers over inner narrow streets is a widespread practice, illegality of extending roofs from building facades for fear of permanent encroachment on public space were highlighted during these consultations. Moreover, the reality of informal areas and their deteriorated physical conditions have mandated the attention to some prerequisites before progressing with environmentally friendly interventions. The deterioration of the sewage system in the case study serves as an example to possible deterioration within similar areas that require ample attention in order to render related adaptation or mitigation interventions effective or sustainable.

The developed OSEPI model relied on empirical inquiry and field work. The identified steps stem from realities of informal areas including physical conditions, local practices, socio-economic aspects as well as governance challenges. Allowing the steps as identified in the OSEPI model, the process attempts to help reach tailored affordable solutions that respond to each local situation. It is vital to note that this model was primarily developed for the low to middle income neighbourhoods within the Cairene context as an example for Mega cities in the Global South. Moreover, it takes into consideration the nature of urban informality that prevails in such cities manifested in a bottom up, incremental development.

In addition, the OSEPI model considers that private territorialities dominate over the public space and there is an ambiguity regarding circulation, recreation and gathering with a lack of formal definition for space use [42]. These limited spaces are always contested among various users. However, they manage to informally organize the space in a temporal mode that would allow different uses at various times catering for various needs. The specific characteristics of location, climate, urban structure, construction materials and methods, socio-culture, and economy are among the aspects that shaped the research and its outcomes. Similar contexts can draw lessons from this study and its outputs with adequate tailoring.

The economic challenges that the project has faced are a common issue within cities of the Global South. The limited available resources for adaptation interventions have been reported by many stakeholders. Since its inception in 2019, the Giza Adaptation strategy has not been able to become implemented due to lack of both adequate funding and local capacities [43]. Current COPs, especially the upcoming COP27 in Egypt, are trying to promote international commitments towards financing adaptation interventions as until now they have not received ample attention. In contrast with adaptation interventions, in most cases, mitigation strategies and interventions have a market that is growing with increased feasibility.

Additionally, the team have produced a handbook [44] outlining the possible tools that can be used by various stakeholders to improve urban microclimate in similar contexts. Moreover, it is to be used as a reference in upgrading projects facilitating the development of more environmentally responsive spaces, while enriching educational and practical domains. The handbook includes sections regarding: Urban design principles, Sun protection and natural ventilation, Building façade and public space, Grey water treatment, Plantation, Green roofs and walls, Street furniture and Community involvement.

## 7. Conclusions

Throughout our research, the team has continuously faced inaccessibility and scarcity of information, especially for the informal sector of the city, hence, we tried to overcome this obstacle with extensive fieldwork. A profound participatory approach was followed in decision making and consultation of various stakeholders to develop and refine the interventions. The process involved intensive participation of local CBOs and area residents. This was led by a group of students under the supervision of the research team, which helped to disseminate the project idea and improve their awareness regarding both environmental issues as well as communities in informal areas. The launch of the pilot project has revealed the local problems that need to be addressed before being able to implement the target interventions, and hence, inspired the research team about the practical steps that must be considered in such contexts which eventually helped developing the OSEPI model. It has shown the complexity of trying to intervene for the purpose of environmental improvements within a deteriorated urban context of informal areas. This adds to the argument of the need for an integrated approach when addressing urban upgrading interventions. A model was proposed to improve the environmental performance in the informal areas, as an attempt for an integrated approach that would consider basic needs as well as the right to a better quality in urban spaces within the risks and implications of climate change.

The work has raised the flag for the current lack of municipal capacities to address climate change related interventions on the urban scale. There is a profound gap between the international goals and agreements such as the SDGs and the Paris agreement, on one hand, and local databases, on the other. Furthermore, there is an extreme lack of integration between data related to climate change, urban climate with related urban heat island effects and urban development plans and actions. The gap extends to implementation capacities and lack of funds to cope with the requirements of the New Urban Agenda and creating a sustainable, resilient, and inclusive city. This is accentuated within a megacity such as Cairo especially within its most vulnerable districts of urban informality. The unstable economic

and political situations in many countries of the Global South hinder cities from being able to address climate resilience. This paper has highlighted the importance of addressing these gaps along with associated challenges and the need to integrate environmental improvement interventions within the scope of urban upgrading. This need to mainstream climate change adaptations into upgrading projects is starting to manifest itself within the latest upgrading project spearheaded by GIZ to be implemented in selected informal districts of Greater Cairo Region, a program that started in January 2019 and still ongoing.

**Author Contributions:** Conceptualization, H.A.E.E.K., A.I. and N.E.; data curation, H.A.E.E.K. and N.M.; formal analysis, H.A.E.E.K., A.I, N.E. and N.M.; funding acquisition, H.A.E.E.K.; investigation, H.A.E.E.K., A.I. and N.M.; methodology, H.A.E.E.K. and N.E.; project administration, H.A.E.E.K.; resources, H.A.E.E.K.; software, H.A.E.E.K.; supervision, H.A.E.E.K. and N.M.; validation, H.A.E.E.K.; visualization, H.A.E.E.K.; writing—original draft, H.A.E.E.K.; writing—review and editing, H.A.E.E.K. and N.M. All authors have read and agreed to the published version of the manuscript.

**Funding:** This publication is part of the German Egyptian Research Fund GERF research project titled: Improving Environmental Performance in Informal Areas and Reducing Urban Heat Islands Phenomenon funded by STDF on the Egyptian behalf (no. 5124) and BMBF on the German behalf.

**Institutional Review Board Statement:** Not applicable.

**Informed Consent Statement:** Not applicable.

**Acknowledgments:** The authors want to thank their German counterparts, namely: Kosta Mathey, Wolfgang Dickhaut, Lutz Katzschner and Christoph Hesse for their valuable contribution in the research. Furthermore, the authors want to thank students' teams from Faculty of Engineering, Cairo University who worked on the research project through courses ARC471: Community Development, Spring 2016 and ARCN335: Community Design and Social Development, Spring 2016, in addition to the group of interns who continued the work during Summer and Fall 2016.

**Conflicts of Interest:** The authors declare no conflict of interest.

## Abbreviations

| | |
|---|---|
| CBO | Community Based Organization |
| COP | Convention of Parties |
| GIZ | German Development Cooperation |
| LST | Land Surface Temperature |
| MENA region | Middle East and North Africa region |
| OSEPA tool | Outdoor Spaces Environmental Performance Assessment tool |
| OSEPI model | Outdoor Spaces Environmental Performance Improvement model |
| PET | Physiological Equivalent Temperature |
| RH | Relative Humidity |
| SDGs | Sustainable Development Goals |
| SUF | Slum Urban Form |
| SUHI | Surface Urban Heat Island |
| UHI | Urban Heat Island |
| UN DESA | UN Department of Economic and Social Affairs |

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
