# Peer review of "Enhancing Livability in Informal Areas: A Participatory Approach to Improve Urban Microclimate in Outdoor Spaces"

_sustainability, doi:10.3390/su14116395_

Round 1

Reviewer 1 Report

Dear authors,

first of all, I would like to congratulate you for your nice work and interesting manuscript dealing with a very important and appealing research topic. Overall, my review has a positive feedback and your paper merits publication, however, please consider some minor revisions beforehand on the list of the points below:

  • Please be sure that you consult in an accurate and careful way the journal's guidelines. For instance, at your title the first letter should be in capital, etc., same for the headings, sub-headings, the authors' names, etc. Same for your reference list. Please read carefully the guidelines.
  • It is strongly suggested to include your keywords in alphabetical order.
  • Line 33: please remove one of the two full stops. 
  • Lines 33-34: please be specific on your argument and cite the author(s)' names. It is recommended to avoid generalities, such as 'another study', be concrete.
  • Line 35: 'megacities, with': please remove the come between the two words.
  • Line 38: please add a comma after 'although'
  • Line 39: please consider the previous comment about being concrete on your citations, mention the author's name of the study.
  • Line 41: please add a comma before 'where'
  • Line 42: please cite Baklanov et al. according to the jounal's template instructions.
  • Line 45: 'with such A context' (grammar error)
  • Line 47: please reform your phrase by avoiding repeating the words 'mitigation and adaptation strategies'
  • Lines 54-58: please reform the phrases to avoid the same style of 'This means...', 'this is...', etc.
  • Line 60: please cite Ningrum (2018) according to the journal's template instructions.
  • Line 62: please add a coma before 'who'
  • Line 66: please add a coma before 'which'
  • It is strongly recommended to clearly mention your research objectives/questions at the end of your introductive part as well as to include a work plan of your research to enable the lecture.
  • Lines 141-143: please check formatting to be in accordance with the rest of the manuscript.
  • It would be more interesting to include your figures after their cross-reference of your text and not too 'far' from their description (example of Figure 2, etc.)
  • Please give more emphasis on the explanation of your methodological part, in its actual version is quite weak. 
  • It is strongly proposed to include at the end of your manuscript a table of nomenclature with all the acronyms appeared. 
  • The paper is well and adequately referenced but the list is not in accordance to the journal's template.

Thank you in advance for your consideration.

Good luck with your manuscript.

Best,

Reviewer 2 Report

The topic of the article is undoubtedly interesting, as the issue of microclimate in outdoor spaces concerns a large percentage of the world's population. This article investigates microclimate in informal areas of Cairo and how to improve it through low-tech interventions.

The authors present selected typical examples and possible specific measures to improve the microclimate. The authors also mention and describe issues related to the attitudes of citizens and institutions in the areas where appropriate measures are proposed.

After studying the article, it can be stated that it summarizes the results of research and proposed measures in an area that significantly affects the living conditions of a large population. Therefore, I evaluate the content of the article positively, I recommend making some formal modifications or additions.

For readers, it would be appropriate to add a list of abbreviations used at the end of the article.
If the figures are not taken over from other authors, i.e. the sources are not cited for them, it is unnecessary to state "authors" or "research team" in the description of the figures.
The formatting of the article generally needs to be adjusted according to the template.

Round 2

Reviewer 2 Report

Dear authors,
the article has been modified and I no longer have any substantial comments on it. I wish you success in further research work in this socially beneficial field.